# Perspectives on Potential Fatty Acid Modulations of Motility Associated Human Sperm Ion Channels

**DOI:** 10.3390/ijms23073718

**Published:** 2022-03-28

**Authors:** Akila Cooray, Ji Hyun Kim, Mee Ree Chae, Sungwon Lee, Kyu Pil Lee

**Affiliations:** 1Department of Physiology, College of Veterinary Medicine, Chungnam National University, Daejeon 34134, Korea; akiladushyantha@gmail.com (A.C.); jihyun@cnu.ac.kr (J.H.K.); 2Department of Urology, Samsung Medical Center, Samsung Biomedical Research Institute, Sungkyunkwan University School of Medicine, Seoul 06351, Korea; meeree11@hanmail.net

**Keywords:** fatty acid, spermatozoa, ion channel, motility, ion homeostasis

## Abstract

Human spermatozoan ion channels are specifically distributed in the spermatozoan membrane, contribute to sperm motility, and are associated with male reproductive abnormalities. Calcium, potassium, protons, sodium, and chloride are the main ions that are regulated across this membrane, and their intracellular concentrations are crucial for sperm motility. Fatty acids (FAs) affect sperm quality parameters, reproductive pathologies, male fertility, and regulate ion channel functions in other cells. However, to date the literature is insufficient to draw any conclusions regarding the effects of FAs on human spermatozoan ion channels. Here, we aimed to discern the possible effects of FAs on spermatozoan ion channels and direct guidance for future research. After investigating the effects of FAs on characteristics related to human spermatozoan motility, reproductive pathologies, and the modulation of similar ion channels in other cells by FAs, we extrapolated polyunsaturated FAs (PUFAs) to have the highest potency in modulating sperm ion channels to increase sperm motility. Of the PUFAs, the ω-3 unsaturated fatty acids have the greatest effect. We speculate that saturated and monounsaturated FAs will have little to no effect on sperm ion channel activity, though the possible effects could be opposite to those of the PUFAs, considering the differences between FA structure and behavior.

## 1. Introduction

### 1.1. Human Spermatozoa

Spermatozoa are a fascinating group of differentiated cells in a complex microenvironment that are produced by spermatogenesis in the male reproductive tract and undergo several morphological and physiological changes to fertilize an egg. Similar to other mammalian spermatozoa, human spermatozoa contain two parts: the head and the flagellum. The head is also divided into two parts: the anterior head, which includes the head with the acrosome, and the posterior head, which is the postacrosomal region [1]. The motile tail of a sperm is a long flagellum of which the central axoneme starts and extends just after the nucleus region. The flagellar movement is driven by dynein motor proteins, which use energy from ATP hydrolysis. This process is tightly regulated by a number of both external and internal factors of the sperm [2,3]. In vitro studies have shown that the human spermatozoan head is a key structural part for successful fertilization, as sperm head shape, acrosomal size, and other parameters affect the sperm’s function [4]. Other parts such as the midpiece [5] and tail of the spermatozoon are as equally important as its head for its proper function. The importance of the ultrastructural regulation of parts of the sperm is reflected by studies that found defects in those parts to be associated with male infertility and related pathologies and abnormalities [6].

### 1.2. Sperm Capacitation

Human spermatozoa undergo a series of events before fertilization, including spermatogenesis, maturation in the epididymis, capacitation, and acrosomal exocytosis. The capacitation and acrosomal reactions occur in the female reproductive tract. Ejaculated sperm resides in the female genital tract for a few hours to become functionally competent. This involves multiple biochemical and physiological modifications, collectively known as sperm capacitation. The endpoint of the capacitation process is the hyperactivation of sperm, which is marked by increased motility [1]. Ion channels that are responsible for the hyperactivation of sperm, and thereby their high motility, are found in the sperm head, neck, and flagellum [7].

During capacitation, the membrane of the sperm head undergoes a series of biochemical modifications to prepare sperm for the acrosome reaction upon reaching the zona pellucida of the egg. This results in the localized activation of sperm-releasing acrosomal enzymes [8]. This event has also been reported to be regulated by ion channels [9]. 

In light of recent discoveries and the introduction of more accurate and rapid techniques, scientists are working to mitigate issues arising from reproductive pathologies. Spermatozoa play a major role, as approximately 50% of infertility cases are reported to be associated with men and abnormalities associated with the male reproductive system. Although there is a high prevalence of male infertility, our understanding of the underlying etiology is currently rudimentary, and further detailed studies are needed to establish more efficient therapies. Current management and therapeutic strategies for male reproductive abnormalities include sperm quality improvement (antioxidants such as glutathione, lycopene, and vitamin E and sperm vitalizers such as carnitine and coenzyme Q10), gene therapy, hormone replacement therapy (androgens, gonadotropins, aromatase inhibitors, and antiestrogens), surgery, assisted reproductive technology (ART), stem cell therapy, and spermatogonial stem cell (SSC) transplantation, depending on the severity and type of abnormality [10,11].

Sperm ion channel function is a ubiquitous factor that contributes to sperm motility. Many fatty acids (FAs) are associated with spermatozoan structure, physiology, microenvironment, and endogenous pathways. However, to date, no studies have investigated the direct effects of FAs and lipids on sperm motility-related ion channels in human spermatozoa. We believe that a proper understanding of ion channel regulation by exogenous factors, such as FAs, can determine their suitability and applicability as therapeutic targets to ameliorate sperm quality. Therefore, we concisely review the available information on ion channels associated with human sperm, the significance of different lipids and FAs on spermatozoa, and finally, we speculate possible directions for future studies on the effect of FAs on sperm ion channels based on the available data on ion channels of other cells that may show relevance in the context of human sperm motility.

## 2. Motility-Associated Major Ion Channels

### 2.1. Calcium Channels

Ca^2+^ is a common secondary messenger that is important in sperm cell motility, capacitation, and the acrosome reaction [12]. Major functions of Ca^2+^ in spermatozoan motility include the activation of the sAC/cAMP/PKA pathway, ATP production, and the maintenance of sperm mitochondrial activity [7,13].

To date, four main calcium channels have been identified to be associated with sperm motility, some of which have been studied extensively, while a few have yet to be defined. These four channels are the cation channel of sperm (CatSper), voltage-gated Ca^2+^ channel (VGCC), transient receptor potential vanilloid (TRPV), and store-operated Ca^2+^ channel (SOCC) [7].

An increase in intracellular Ca^2+^ ion concentration is essential for changes in flagellar function in the transition of sperm from the activated form to its hyperactivated form, where the sperm-specific Ca^2+^-permeable channel CatSper plays a major role [14]. CatSper is the most studied sperm Ca^2+^ channel because it is highly specific to sperm and plays a significant role in flagellum hyperactivation. In sperm of tested animals, such as mice and bovines, intracellular alkalinization seems to be the key regulator of the ion channel. However, the regulation of CatSper in humans and other primates is more complex because the ion channel can be activated by a wide range of agonists such as progesterone, egg coat proteins, and albumin, despite the ion channel being pH sensitive [9]. Ca^2+^-dependent sperm motility is mainly achieved through CatSper-mediated Ca^2+^ influx, which propagates through the midpiece and head upon channel activation [15]. It can also occur through the mobilization of stored calcium ions from the organelles at the sperm midpiece and other calcium channels [16]. Highlighting the significance of the ion channel, inhibition of CatSper in mature human spermatozoa by NNC 55-0396 (NNC) resulted in a significant decrease in sperm viability, percentage of motile sperms, curvilinear velocity, and other motility associated parameters [17].

VGCCs are a group of membrane-bound channels that mediate Ca^2+^ influx into sperm cells [18]. The activities of L-type VGCCs have been reported in the motility of human sperms cells [19] and compounds known to inhibit the channels have shown a decrease in human sperm motility [13]. However, the complete elucidation of the mechanism by which the VGCCs contribute to sperm mortality requires further investigation.

TRVPs are another group of cation channels that may contribute to human sperm cell motility. There are six subtypes, consisting of TRPV1–6. TRPV1–4 are thermosensitive and moderately permeable to Ca^2+^, whereas TRPV5 and TRPV6 are not thermosensitive but highly selective for Ca^2+^ [20,21]. Of the subtypes of TRPV, TRPV1 has been reported to be positively correlated with sperm motility, and sperm migration towards a temperature gradient of 31–37 °C has shown higher expression of TRPV1 mRNA and proteins, confirming its function in motility [22]. The significance of the TRPV1 channel has been more extensively studied in other animals [23,24,25] such as zebrafish [24] and bulls [25], and suppression of the ion channel results in decreased sperm motility. In *Labeo rohita*, the activation of the ion channel by external factors such as N-arachidonoyl dopamine increased the duration of sperm motility [23]. However, the implications made based on the above animal studies should be carefully transferred to human sperm cells, as data on TRPV1 regarding human sperm cell motility is scarce. TRPV4 is another subtype of the TRPV channel family that has been shown to have functional significance in human sperm motility. The literature indicates that the sperm of the swim-up fractions displayed an increased level of TRPV4 expression compared to that of the non-motile sperm [26]. Furthermore, Mundt et al. claimed that the channel could be involved in human sperm motility and plays a central role in gating other sperm motility-related channels, such as CatSper channels [27].

SOCCs are a group of voltage-independent channels in the plasma membrane. They are activated by the intracellular binding of Ca^2+^ to the channel. The literature indicates that SOCC inhibitors reduce asymmetrical flagger beating and turning movements of ascidian spermatozoa, while blocking SOCCs in human sperm strongly inhibits their motility, highlighting the functional significance of the channel in motility [28,29].

In addition to the four main calcium channels mentioned above, several other calcium channels have been reported that are less evident and more poorly defined. Similar to SOCCs, Na^+^/Ca^2+^ exchangers are voltage-independent ion channels that are indispensable for human sperm motility. The inhibition of these channels by substances such as KB-R7943 reduced the motility of human sperm [29,30]. Cyclic nucleotide-gated channels have also been proposed to mediate Ca^2+^ influx in the sperm, and may indirectly contribute to sperm motility, as they might activate CatSper via a PKA-dependent pathway [9]. Further studies are required to validate this claim. 

### 2.2. Potassium Channels

Potassium channels regulate membrane potential and sperm cell motility. They assist sperm motility during the late stages of capacitation, especially during membrane hyperpolarization [31]. Several potassium channels have been observed to be important for sperm motility across many animal species [7]. They are inwardly rectifying K^+^ channels (Kir) [32], voltage-gated potassium channels (K_V_ 1.1) [33], cyclic nucleotide-gated K^+^ channels (CNGK channels) [34], and Slo K^+^ channels [31,35,36]. These channels show variance across different animal species, where CNGK is mainly reported to be found in aquatic animals [34]. In human sperm, Slo1 and Slo3 are shown to play an essential role in motility. However, there is controversy among the findings regarding the functional significance of Slo K^+^ channels and the principal K^+^ channel in humans. Mannowetz et al. suggested that Slo1 is the principal potassium channel of human sperm, which is found in other excitable tissues. It can detect changes in both voltage and intracellular Ca^2+^ concentrations, as the channel has two K ^+^ conducting regulators that have binding affinity to Ca^2+^ ions. K^+^ currents are dependent on intracellular Ca^2+^, whereas they are independent of intracellular pH changes [31]. Furthermore, the K^+^ currents were affected by Slo1 channel blockers. This confirms the functional non-redundancy of the Slo1 channel in human sperm, thereby contributing to sperm motility [31].

Sol3 is only found in male germ cells [31]. Compared to Slo1, Slo3 is largely activated by intracellular Ca^2+^, where it is modestly sensitive to intracellular pH changes. Based on a study by Brenker et al. [35], Slo3 was identified as the principal K^+^ channel in human sperm. Slo3 is responsible for controlling the outward K^+^ ion flow inhuman sperm, which suggests that the function of the ion channel would be to provide negative feedback during hyperpolarization, thereby decreasing the open probability of CatSper channels and reducing Ca^2+^ influx [35]. Based on the available literature, it is unequivocal that potassium channels are essential for human sperm motility. However, further work is needed to establish the functional identity of each K^+^ channel found in human sperm and their relevance to sperm motility.

### 2.3. Proton Channels

The development of hyperactivated motility of human sperm depends on the intracellular pH. The main regulator of sperm intracellular pH during capacitation and hyperactivation is the voltage gated proton channel Hv1 [37]. The activity of Hv1 is directly linked to the Ca^2+^ influx of the sperm by activating the pH-dependent CatSper channels. This Hv1 dependent calcium ion channel regulation leads to concomitant changes in the motility patterns of human sperm [38]. The Hv1 channel is mainly localized in the principal region of the flagellum. Subcellular localization of the proton ion channel is consistent with sperm motility and association with other ion channels participating in motility [39]. Although the main pH regulator of sperms is Hv1, it is possible that there could be non-electrogenic transporters that may participate in proton extrusion of sperm, which cannot be identified by current techniques such as patch clamping [37].

However, seminal plasma with concentrations of zinc as high as 3 mM inhibits premature activation of freshly ejaculated sperm by inhibiting proton channel activity. Once ion channel inhibition is restored by zinc–albumin chelation and absorption by the female reproductive tract, the sperm are capacitated and highly motile, indicating the significance of the Hv1 channel in sperm motility [9,37]. Activated Hv1 channels were also seen in reports by Keshtgar et al., where the inhibition of human Hv1 resulted in low sperm viability, percentage of motile sperm, curvilinear velocity, and other motility associated parameters. These observations were similar to those of the CatSper channel [17].

### 2.4. Sodium Channels

Research data hitherto suggest that sodium channels of sperm play a pivotal role in fertilization, especially through the regulation of sperm motility, as these channels are essential in regulating membrane potential. Two types of Na^+^ channels were found on the spermatozoa membranes: the epithelial sodium channel (ENaC) and the voltage-gated sodium channel (VGNCs/Na_v_) [40].

The VGNC family consists of nine members named Na_v_1.1–Na_v_1.9, which share more than a 50 percent amino acid identity. All members of VGNCs are found in the sperm membrane with regional specificity, implicating their specific signaling and metabolic pathways in spermatozoa. However, there is a paucity of information to date regarding the functional specificity of these members [7]. In relation to sperm motility, Na_v_1.8 has been widely studied. Data from Cejudo-Roman et al. show that Na_v_ channels such as Na_v_1.8 are involved in sperm progressive motility rather than in the hyperactivation of sperms. Furthermore, several studies have shown increased progressive motility and protein tyrosine phosphorylation in the presence of VGNC activators such as veratridine [40,41,42].

The ENaC is composed of four subunits, α, β, γ, and δ; only α and δ subunits are found in mammalian sperm ENaCs. The subcellular localization of these ion channels in the human sperm is limited to the flagellar midpiece. A study conducted by Kong et al. suggested that ENaCs in sperm could affect the motility of human sperm. The results of this study indicated that inhibition of the ion channels significantly increased sperm motility in both normozoospermic and asthenospermic donors [43].

### 2.5. Chloride Channels 

Information regarding chloride channels and FAs is scarce. Chloride channels are important for sperm motility, as they function within many events, such as plasma membrane hyperpolarization, cAMP/PKA-induced protein phosphorylation, and alkalization of the sperm cytosol, all of which are associated with sperm capacitation [44]. To date, three chloride channels have been reported to be involved in mammalian sperm motility: cystic fibrosis transmembrane conductance regulator (CFTR), Ca^2+^-activated chloride channels (CaCC), and chloride channel-3 (ClC3) [7].

ClC3 is the most studied member of these channels. These ion channels are found in the flagellum, acrosome, and midpiece [45,46]. Liu et al. claimed that the chloride ion channel is a key factor in regulating sperm motility. According to the authors, the involvement of the ion channel in sperm motility was further confirmed, as the asthenozoospermic population had a lower expression of ClC3 than the normozoospermic population [45]. This hypothesis is further supported by the earlier studies by Smith et al., which showed the potential binding of the ClC3 ion channel to a key enzyme involved in the regulation of sperm motility: PP1γ2 [47]. However, the exact mechanism underlying ion channel involvement has not been fully elucidated.

Similarly, CaCCs and CFTRs were found to affect sperm motility in different species, such as guinea pigs [48] and mice [49]. It can be predicted that ion channels may have the same effect on human sperm because of the high sequence identity between the studied species and humans. Nevertheless, further studies are required to describe the functional relevance of these ion channels in the context of human sperm motility.

## 3. Lipids and Spermatozoa

Lipids in the spermatozoon plasma membrane are essential for sperm function and integrity. The metabolism of lipids is highly regulated in sperm to ensure its physiological processes, which include sperm motility, capacitation, and acrosome reaction [50,51]. The lipid composition of mammalian spermatozoa is different from that of other cells because sperms have a high amount of neutral lipids (diacylglycerols) [50]. The composition of the head and tail are different: the plasma membrane of the head shows lateral heterogeneity regarding the surface molecular topology [52], while lipids show a unique asymmetrical distribution in the bilayer [53]. Shan et al. reviewed different types of lipids of the mammalian spermatozoan plasma membrane and their deficiency-associated pathologies in fertility, including sperm motility [50]. Nevertheless, predictions of the effects of lipids in human sperm motility should be carefully made, as there is limited data available in this regard.

### 3.1. Fatty Acid Composition of Spermatozoa and Seminal Fluid

Lipids, a broad class of molecules, contain FAs as a part of their structure. FAs are generally long hydrocarbon chains consisting of a carboxylic acid group at one end, and they play multiple biological roles as either single molecules or as components of molecules. These roles include participation in cell membrane structure, supplying energy for metabolic processes, and as signaling molecules. FAs can be categorized into groups based on the number of double bonds: saturated FA (SFA)—no double bonds, monounsaturated FA (MUFA)—having one double bond, and polyunsaturated FA (PUFA)—having more than one double bond [54,55]. Human spermatozoa consist of approximately 30 different FAs, including SFA, MUFA, and PUFA (n-6 and n-3) [56]. FAs of spermatozoa and seminal fluid mainly consist of palmitic acid, stearic acid of SFAs, oleic acid of MUFAs, docosahexaenoic acid (DHA) of n-3 PUFAs, and arachidonic acid (AA) and linoleic acid of n-6 PUFAs. FA composition by category is represented in detail by Collodel et al. [57] and the structures of major FAs found in spermatozoa and seminal plasma are shown in Figure 1. According to Zerbinati et al., there is a difference between the FA composition of human spermatozoa and seminal plasma of normozoospermic samples. DHA levels were approximately six times higher in spermatozoa than in seminal plasma. Additionally, myristic, palmitic, palmitoleic, linoleic, vaccenic, eicosadienoic, dihomo-γ-linolenic, and docosapentaenoic acids were higher in spermatozoa than in seminal fluid. However, behenic, lignoceric, oleic, and mead acids were higher in seminal plasma [58].

FA concentration shows regional specificity in sperm, where the FA profiles are different between the head and tail of the sperm. A higher concentration of n-3 PUFAs, especially DHA, was measured in the human sperm head region in comparison to the tail region [56]. This may be attributed to the function of the head region, as it is critical for acrosome biogenesis. In addition, the concentration of SFA, MUFA, and total ω3 FAs was significantly higher in the human sperm head than in its tail [56].

### 3.2. Fatty Acids Associated Sperm Abnormalities

Most essential FAs related to spermatozoa and the reproductive tract are not synthesized in the human body. Hence, dietary intake of these FAs is needed. Several studies have shed light on the association between diet and sperm quality parameters [59], where dietary patterns, such as the Mediterranean diet, may improve semen quality [60]. 

Low dairy fat intake is associated with high progressive motility and normal morphology [61,62], whereas a high intake of dairy fat has shown a negative relationship with sperm count and concentration [63].

Considering the SFAs, low sperm count and sperm concentration were found in men with a high dietary intake of SFAs [64]. Adding to this, a more recent study done by Eslamian et al., which compared large groups of normozoospermic and asthenozoospermic participants, revealed that a high level of palmitic and stearic acid intake was associated with asthenozoospermia [65].

Of PUFAs, a high intake of omega-3 FAs such as DHA has resulted in a low probability of the person being asthenozoospermic [66]. Omega-3 FAs have also been shown to improve normal sperm morphology, semen volume, and other sperm quality markers [64,67]. On the other hand, trans PUFAs show a negative relationship with sperm quality, where low sperm count was seen in men with high trans FA intake [68]. Furthermore, asthenozoospermic subjects were also reported to have a high trans FA in their diet as compared to normozoospermic subjects [65].

### 3.3. Normozoospermic and Non-Normozoospermic Spermatozoa FA Profiles

Various studies have suggested that lipids and FAs play vital roles in sperm structure and physiology, including viability and events during fertilization. This is reflected in the different FA profiles observed in the spermatozoa of normozoospermic and non-normozoospermic men [58]. 

High levels of SFAs such as stearic acid and palmitic acid [69] and low levels of DHA, PUFA, total ω3 FA, and ω3 PUFA have been observed in the FA profiles of asthenozoospermic spermatozoa when compared to those of normozoospermic spermatozoa [56,67]. According to Aksoy et al., oligozoospermic sperms show a relatively lower level of DHA than normal sperms [70].

### 3.4. Fertile and Infertile Subjects’ Spermatozoa FA Profiles

Defective sperm function is a primary cause of infertility and subfertility. The FA profiles of spermatozoa vary between patients and fertile men. Among SFAs, increased levels of palmitic acid were observed in infertile men than in healthy men [71]. Stearic acid was higher in subfertile [56,70] and lower in infertile patients [71], when compared to fertile individuals.

To date, PUFAs have been demonstrated to increase oleic acid [70] and decrease DHA [70,72] among subfertile patients. Higher concentrations of linoleic acid and AA were observed in infertile patients [73]. However, other PUFAs, such as DHA, were lower in infertile men [71,73].

### 3.5. Sperm FA Metabolism and Modifications

Along with the dietary FA intake and the FA profiles of spermatozoa and seminal fluid, FA metabolism and modification of sperm membranes have been reported to influence male fertility and sperm membrane stability, capacitation, and immaturity [57]. Involvement of FA metabolism in sperm function is also seen in the subcellular proteomic data presented by Amara et al. [3]. This study revealed that the majority of the metabolic proteome of sperm is involved in lipid metabolism. This included the identification of various peroxisomal proteins known to participate in the oxidation of very-long-chain FAs. The inhibition of lipid metabolism also affects sperm motility. Ferreira et al. provided further evidence to support the importance of lipid and FA metabolism in the sperm membrane. The study concluded that the addition of glycerophospholipids to the sperm membrane could improve sperm viability, motility, and resistance to oxidizing agents such as hydrogen peroxide [74].

Collodel et al. reported several studies indicating the significance of FA metabolism in sperm maturation and quality, and male reproductive pathologies such as varicocele, globozoospermia, and infertility. These studies used F_2_-isoprostane (a secondary marker of oxidative damage) and phospholipase A2 (PLA2: the main phospholipase in sperm) as indicators of PUFA metabolism. The collectively obtained data suggested that the FA metabolism and the FA composition of the sperm are interrelated parameters, where an increased susceptibility to non-enzymatic FA oxidation was found in globozoospermia and infertile patients with varicocele [75,76,77,78]. Additionally, cis-polyunsaturated FAs, such as AA, have been identified to directly affect reactive oxygen species (ROS) generation in a time- and dose-dependent manner, leading to the peroxidase damage of human spermatozoa and ultimately the loss of motility [79].

## 4. Fatty Acids and Sperm Motility

A study conducted by Zalata et al. including normozoospermic, asthenozoospermic, and oligozoospermic groups showed a significant positive correlation between motile sperm concentration and concentrations of DHA, PUFA, ω3 higher unsaturated FA (with at least 20 carbon atoms and three double bonds), and total ω3 FA [56]. Furthermore, in the same study, the aforementioned FA resulted in a significant positive correlation with the linear velocity of the sperm. In an earlier study, Nissen et al. showed the same effect of DHA on sperm motility [80]. Although there is evidence of several dietary FA affecting sperm motility, the data available at present are insufficient to draw a conclusion between the direct effect of the human sperm FA profile on its motility (reviewed in Esmaeili et al. [81]). However, we presume the dietary FA which increased sperm motility may have reflected on their increased spermatozoa FA profiles, as they coincide with the differences obtained from defective sperms and non-fertile men. This hypothesis is further supported by a study conducted by Conquer et al., which demonstrated an increased DHA status and sperm motility in an asthenozoospermic group following a three month DHA supplementation [82].

FAs have also been shown to have inhibitory effects on human sperm motility. SFAs, such as palmitic acid and stearic acid, demonstrated little to no inhibitory effect on sperm motility, whereas unsaturated FAs like oleic acid [76,83], linoleic acid, and linolenic acid, inhibited sperm motility. Among the selected FAs, the degree of inhibition was observed for oleic acid, linoleic acid, and linolenic acid was in the following order: oleic acid < linoleic acid < linolenic acid [83].

### 4.1. Effect of Lipids/FAs on Ion Channels

Despite the unequivocal importance of FAs in sperm physiology, investigations of the direct effects of FAs on human spermatozoa motility are ongoing. Hence, here we summarize the data available on FAs and similar ion channel activity found in other cells and animal models, which may reflect sperm motility. We summarized the available data on FAs modulating similar ion channels in other cells in Table 1. We believe this will guide meticulous research in the area that will generate more detailed descriptions and will reveal currently unnoticed attributes of FAs and the regulation of human sperm motility-related ion channels.

#### 4.1.1. Calcium Channels and FAs

Since CatSper is a sperm-specific cation channel and the importance of free fatty acids (FFAs) on sperm ion channels is relatively unexplored, literature is unavailable to draw any conclusions regarding the effect of FAs on the CatSper channel. However, few studies have shown the effect of steroids such as progesterone on ion channel activity and support the claim that steroids act agonistically on human CatSper [117,118,119,120].

Considering other calcium channels, FAs elicited different effects on different Ca^2+^ ion channels. Evidence from VGCCs in animal cells shows an inhibitory effect of PUFAs on VGCCs. Micromolar concentrations of PUFAs (i.e, eicosapentaenoic acid (EPA), DHA, AA, linolenic acid, linoleic acid, conjugated linoleic acid [86]) were able to antagonize the channel activity with approximately 8µM of AA, resulting in a 50% reduction of the L-type Ca^2+^ channel current in rat ventricular myocytes [87]. The same authors reported that MUFAs (oleic acid) and SFA (stearic acid) had no ostensible effect on VGCC under the same settings. However, PUFAs showed a stimulatory effect on TRPV channels, where AA activated both TRPA1 [89] and TRPV4 [90] in human embryonic kidney cells (HEK) cells.

#### 4.1.2. K^+^ Channels and FAs

The modulation of K^+^ channels by FAs has been documented by numerous instances in different tissues and cells. Despite decades of work on FAs, spermatozoa, and K^+^ channels separately, it is noteworthy that we did not find reports related to FAs and K^+^ ion channels that are specific to sperm; therefore, this section recapitulates the arguments in favor of human sperm K^+^ ion channels, which are also found in other cells.

Findings related to FAs with K_v_ channels and Slo1 channels in other cells may help to complete the puzzle to solve their FA associated regulatory mechanism in human sperm cells.

The extracellular application of long-chain PUFAs (AA, cis-linoleic acid, and DHA) inhibited the K_v_ current of many animal cells, including bovine zona fasciculata cells [96] and mouse and rat cardiomyocytes [97], whereas trans-PUFAs (linoleladiac acid) and SFAs (stearic acid) failed to inhibit the ion channel under similar conditions. The current reduction of K_v_ channels by PUFAs was as high as 68% in the presence of AA on the rat K4.2 channel [121]. FFAs showed similar results in K_v_ in human cells. AAs act as an open channel blocker for K_v_ in human cardiac cells, where they have the ability to directly interact with multiple amino acids located in the pore domain of the channel [98]. The different results exerted on K_v_ by FFA could be due to the FA’s lipoelectric mechanism (the charge of the head group determines the direction of the effect) and its involvement in several molecular rearrangement steps leading to ion channel regulation [122]. In addition to this, Elinder et al. [123] reviewed different effects of PUFAs on different types of voltage-gated potassium channels, where some of the potassium channels (K_V_7, K_v_10–12, and K_ca_) resulted in an increased amplitude of current in the presence of the PUFAs.

Several studies support the hypothesis that PUFAs stimulate the Slo1 ion channel. It was observed that FAs interact with ion channel proteins directly and specifically in GH3 cells. This claim is supported by findings that Slo1, which is not related to changes in membrane fluidity, and products arising from lipoxygenase, cyclooxygenase, and superoxide were not responsible for altering the ion channel activity [94]. In the same study, cis-free FAs, including palmitoleic, oleic, linoleic, linolenic, and eicosapentaenoic acid (EPA) increased the Slo1 activity of GH3 cells. Interestingly, it was also recorded that at least one cis double bond was required for the FA activation of the channel, and the magnitude of the activation was correlated to the degree of unsaturation of the FA [94,124]. Furthermore, a similar effect was obtained by Martin et al. where 10µM AA increased the open probability of the Slo1 channel of human vascular smooth muscle channel and HEK cells [93]. Further dissection into human Slo1 activation by PUFAs revealed that the stimulatory effect of PUFA (i.e., DHA) can be independent of voltage or Ca^2+^ sensors and is specific to the structure and availability of auxiliary subunits of the channel. Furthermore, molecular determinants, such as the para-group of a Tyr residue near the ion conduction, residues at the N-terminus and transmembrane segments of the auxiliary subunits, and double bonds present at the halfway position, are found to be critical in activating hSlo1 [125,126,127]. Other notable features of hSlo1 and FFAs are that FFAs may show a diverse modulatory repertoire on different cells, as PUFAs regulate different auxiliary subunits of the channel, and that the subunits are expressed in a highly tissue-specific manner [127].

#### 4.1.3. Hv1 and FAs

Although the voltage-gated proton channels lack the conventional ion-conducting pore domain, the voltage sensing of the ion channel is similar to that of other voltage-gated ion channels, with the only difference being that the two voltage sensing domains (VSDs) act together as a dimer [128]. The effect of FAs on Hv1 is rather reminiscent among many cells, but the data are mostly limited to AA. PUFAs such as AA shift the gating voltage in the negative direction and increase the maximal current of the channel in HEK cells [102], many human immune cells including neutrophils [103], and eosinophils [103,105], and murine macrophages [104]. As demonstrated by Kapus et al., the order of potency in stimulating the H^+^ current in murine macrophage cells was AA > palmitoleate = palmitelaidate > linoleate > oleate > elaidate, and SFA did not show any activity against Hv1 [104].

AA increases the proton currents of the tested cells directly [102,103,104,105,106,107] and can also activate the channel in a protein kinase C-dependent manner [106]. Additionally, the activation of Hv1 was dependent on the major cellular lipase, PLA2. The ion channel was not activated in the absence of PLA2, but its activity was fully restored upon exogenous application of AA, indicating the physiological requirement of enzyme-generated AA for the activation of the H+ channel [102,129].

#### 4.1.4. Other Ion Channels and FAs

PUFAs, such as DHA and EPA, showed an inhibitory effect on VGNC in both animal [108,110] and human cells [109,111]. Additionally, an application of AA showed duality on Na_v_ depending on the depolarization potential in the rat Na_v_1.4 channel [112]. When considering the MUFAs and SFA, they had little or no effect on inhibiting both human and animal Na_v_ channels [108,111]. Furthermore, Kang and Leaf demonstrated that the effects of such PUFAs on Na_v_ could be due to the direct binding of the FAs to ion channel proteins [130]. ENaC channels are another type of Na^+^ channels that are found in human sperms and associated with sperm motility. ENaC found in rat cortical collecting ducts (CCD) was inhibited by 2µM AA [113].

Among the chloride channels, the CFTR channel was inhibited by AA in baby hamster kidney cells. The authors postulated that AA interacts with the positively charged amino acid side chains in the cytoplasmic vestibules of cells to insert its inhibitory action [116].

In summary, sperm ion channels and their pharmacological modulations are relatively less explored due to many practical difficulties such as the patch clamping technique requiring highly technical skills and that processing a motile cell like sperm is more difficult than other cell types [131]. Among FAs, PUFAs are more potent in modulating ion channels, while trans-FFAs and SFA tend to show little to no activity, if there is any activity, which is usually antagonistic to the effect of PUFAs on the same ion channel. There are several explanations as to why such a greater modulation is seen by PUFAs. Ionic channels can be redox-modulated, PUFAs may serve as electron carriers, and their electron shuttling ability is proportional to the degree of unsaturation. This may reflect the fact that highly unsaturated FAs have a greater effect on ion channels [132]. Another reason is the greater conformational mobility of PUFAs compared to those of SFAs. Among the PUFAs, different FAs may insert different regulatory mechanisms on ion channels depending on FA characteristics (degree of saturation, length of acyl chain, charge of carboxylic group) [124], binding site to the ion channel [123] (intracellular cavity, extracellular entrance to the pore, the intracellular gate, the interface between the extracellular leaflet of the membrane bilayer and its channel domain, and the interface between the leaflet and the pore domain), and channel specific properties. This suggests FAs to be a putative sperm ion channel modulator to overcome sperm motility associated abnormalities. However, when considering dietary FAs as potential ion channel modulators, care should be taken to address the other aspects that may arise from fatty acid metabolism. This includes minimizing oxidative stress (i.e., ROS) generated in the FA metabolism process. Such adverse effects could be overcome by supplementing together with antioxidant vitamins (Vitamin E) [133].

Altogether, FFAs are significant in male fertility and may have the ability to control the motility of human spermatozoa in an ion channel-dependent manner. They also have the potential to be used as therapeutic agents that target ion channels to overcome many male reproductive pathologies. However, further investigations are essential to validate their applicability and elucidate their mechanisms of action.

## Figures and Tables

**Figure 1 ijms-23-03718-f001:**
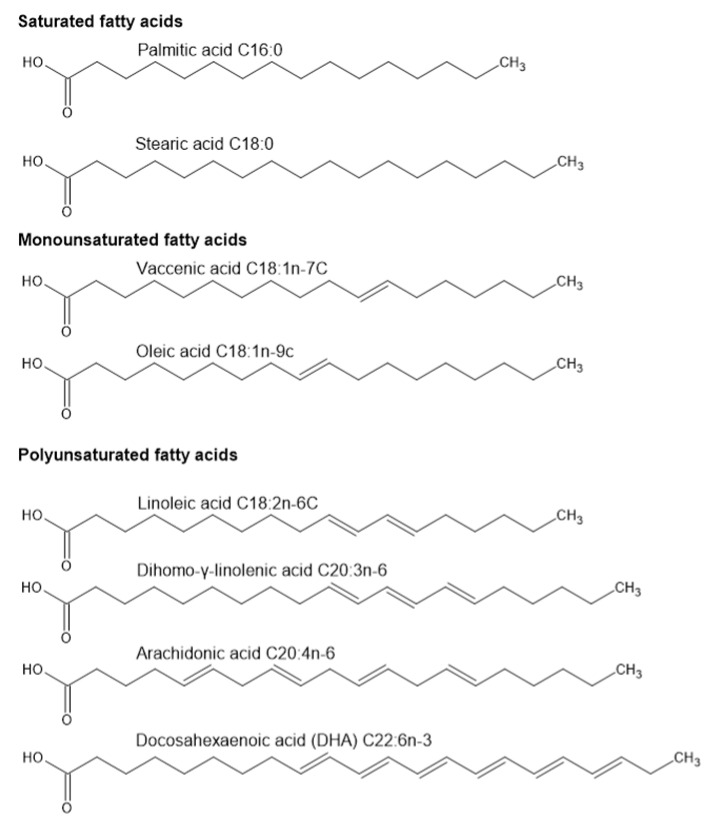
Most common fatty acids found in human spermatozoa and seminal plasma.

**Table 1 ijms-23-03718-t001:** Ion channels and their pharmacological regulation of FFAs.

Ion	Ion Channel	Sub Cellular Localization on Sperm	Function	Fatty Acids/ Lipids on Ion Channels	
				Related Animal Studies	Related Human Studies
Calcium	CatSper (cation channel of sperm)	Principal piece of the flagellum [84]	Calcium influx and activation of Ca-dependent hyperactivated motility [85]	N.A.	N.A.
	Voltage gated Ca^2+^ channel (VGCC)	Neck and principal piece of the flagellum [18]	Mediating Ca^2+^ influx in response to action potential and subthreshold signals participating in sperm acrosome function [18]	0.8 µM and 2.1 µM EPA inhibited the voltage gated L-type Ca^2+^ current by 50% in neonatal rat heart cell and in adult ventricular myocytes accordingly. Other PUFAs (DHA, AA, linolenic acid, linoleic acid, conjugaed linoleic acid, and eicosatetraynoic acid) had similar effects on calcium currents. MUFAs (Oleic acid) and SFA (Stearic acid) had no effect on L-type calcium current. [86]	N.A.
			N.A.	8.5µM of AA caused 50% inhibition of L-type Ca^2+^ channel current in adult rat ventricular myocytes. [87]	N.A.
	Transient receptor potential vanilloid (TRPV)	Sperm head, acrosome, neck and the flagellum, mostly in the flagellum of the human sperms [88]	Regulation of voltage gated Ca^2+^ influx and temperature reception [88]	N.A.	AA activated TRPA1, then ethanolamide or aminoacid/neurotransmitter derivatives in HEK293 expressing hTRPA1 cells. [89]
					AA activates the TRPV4 channel via epoxyeicosatrienoic acids in HEK293 cells. [90]
	Store-operated Ca^2+^ channel (SOCC)	Sperm head, neck and midpiece [91]	Supplying Ca^2+^ from extracellular environment in a voltage independent manner [92]	N.A.	N.A.
Potassium	Slo1	Sperm flagellum [31]	K^+^ efflux of the sperm [31]	N.A.	10 µM arachidonic acid increases the open probability of BKca (Slo1) channel of human vascular smooth muscle cells (VSMCs) and HEK cells in the presence of the β1-subunit. [93]
					cis FFA, palmitoleic, oleic, linoleic, linolenic, EPA increased Slo1 activity in GH3 cells. [94]
	Slo3	Principal piece of the flagellum [7]	K^+^ efflux of the sperm [35]	N.A.	N.A.
	Voltage gated potassium channels	Principal piece of the flagellum [95]	K^+^ efflux [95]	Arachidonic acid (1–20 µM), cis-linoleic acid inhibit the K_v_1.4 current in bovine zona fasciculata cells. Trans-PUFA (linoleladiac acid) and SFA (stearic acid) failed to inhibit the K_v_1.4 current [96].	10 uM AA did not affect the K_v_ activity of VSMCs cells [93].
				Extracellular application of long chain PUFAs (AA, DHA) inhibited the K^+^ current channels (K_v_ 1.5) in mouse and rat cardiomyocytes. [97]	AA acts as an open channel blocker for hK_v_1.5 channel of human cardiac cells. AA itself can interact with multiple amino acids located in the pore domain of the channel. [98]
				Physiological concentrations of DHA and AA (3–10µM) potently and irreversibly inhibited the K_v_ of rat olfactory receptor neurons. [99]	α-linolenic acid (ALA) blocked arterial specific K_v_1.5 protein with Ic50 ~ 3.7µM. EPA and DHA reduced the steady state levels of the ion channel protein, but ALA did not. [100]
				Two outward K^+^ currents of ferret cardiomyocytes were inhibited by EPA and DHA whereas inwardly rectifying K^+^ current was unaffected. MUFAs and SFAs lacked the effect on K^+^ channels. [101]	N.A.
Proton	H_v_1	Principal piece of the flagellum [38]	Proton extrusion resulting intracellular alkalization and activation of spermatozoa [38]	N.A.	AA increased the proton current through mouse H_v_ channel (mH_v_1) in heterologous expressed in HEK293T cells. Application of phospholipase 2 (PLA2) that generates AA from cell membrane phospholipids also stimulated the channel activity. (Similar to approx. 20µM AA) [102]
					AA increased H^+^ selective conductance of human neutrophils [103]
					AA greatly enhanced the slowly activating H^+^ currents of murine macrophages. Effects of AA were not mediated by lipoxygenases (LOX) or cyclooxygenases (COX). The order of potency to stimulate the conductance was: AA > palmitoleate = palmitelaidate > linoleate > oleate > elaidate. Saturated fatty acids were inactive against H_v_1. [104]
					The proton current of human eosinophils were augmented in the presence of 10µM AA. [105]
					AA enhanced the voltage gated proton conductance in human eosinophils. Upon activation by AA, activation was four times faster, H^+^ current amplitude was approximately five times higher, and gated voltage was shifted to more negative voltages. [106]
					Application of exogenous arachidonate can open the H^+^ channel of human cytoplasts in the absence of superoxide generation and the used concentrations of AA did not induce H^+^ permeability in liposomes of the cells. [107]
Sodium	Voltage gated Na^+^ channel (VGNCs,Na_V_)	Principal piece, connecting piece, head, midpiece [41,42]	Maintenance of progressive motility [41,42]	PUFAs (DHA, EPA, linolenic acid, linoleic acid) inhibited the sodium current in dorsal rat ganglion cells in a dose dependent manner. Higher degree of unsaturation was resulted greater inhibition. MUFAs (Oleic acid, palmitoleic acid) and SFAs (Stearic acid, palmitic acid) inhibited the current to a lesser extent than PUFAs did. [108]	Eicosapentaenoic acid (EPA) and docosahexaenoic acid (DHA) reduced the voltage gated Na^+^ current in human atrial myocytes in a concentration dependent manner. IC50 EPA-10.8 µM; DHA-41.2 µM [109]
				Extracellular application of EPA (5 or 10 µM) caused an inhibition of voltage gated sodium currents of neonatal rat ventricular myocytes. The same concentrations of AA caused lesser inhibition. [110]	Both saturated and unsaturated fatty acids inhibited the current via α subunit of the human cardiac Na^+^ channel. But only PUFAs (EPA, DHA) inhibited the complete human myocardial Na^+^ channel. [111]
				AA had both inhibitory and activating effects on rNa_v_1.4 depending on the depolarization potential. At −30 or −40 mV depolarization potential, AA activated the ion channel, but potential over −10 mv significantly inhibited the ion channel activity. [112]	
	Epithelial Na^+^ channel (ENaC)	Flagellar midpiece [43]	Regulation of capacitation-associated hyperpolarization [43]	2µM AA inhibited the ENaC activity by 50% in rat cortical collecting duct (CCD) cells. [113]	
Chloride	Chloride channel-3 (ClC3)	Flagellum, neck, midpiece [46]	Regulation of sperm volume and motility [45]	N.A.	N.A.
	Calcium activated chloride channels (CaCC)	Head [114]	Assist in Ca^2+^ dependent Cl^-^ currents of the sperm [114]	N.A.	N.A.
	Cystic fibrosis transmembrane conductance regulator (CFTR)	Midpiece of the sperm [115]	Efflux of Cl^-^ upon capacitation and mediating HCO3 entry [44,115]	AA inhibited the activity of CFTR chloride channels of baby hamster kidney (BHK) cells. AA may interact with the positively charged amino acid side chains in the cytoplasmic vestibule to block the channel. [116]	N.A.

## Data Availability

Not applicable.

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
