# Peer review of "Perspectives on Potential Fatty Acid Modulations of Motility Associated Human Sperm Ion Channels"

_ijms, 2022, doi:10.3390/ijms23073718_

Round 1

Reviewer 1 Report

Dear authors,

after reading the review entitled "Perspective on Potential Fatty Acid Modulations of Motility Associated Human Sperm Ion Channels" please find my comments :

  • the review is very interesting, on a subject where a lot of work still needs to be done, as clearly stated;
  • The table 1 gives a good overview of the main data regarding the published effects of FAs on ion channels;
  • The authors do not mention the potential risks, when speaking of dietary supplementation with PUFAs, of oxidation, and the need to supplement also with vitamin E, a point that could have impacts on ion channels activity in pathophysiological conditions;
  • The study of sperm ion channels has been slowed down by the technical limitation regarding the use of patch-clamp on these very special cells. This point is not discussed and I believe I it is really an issue on this topic. Several reviews or studies have been published since 2013, either on human or animal spermatozoa: Lishko 2013 (Methods in Enzymology, Volume 525), Mansell 2014 (Molecular Human Reproduction, Vol.20, No.5 pp. 392–408, 2014) or Chung 2017 (Elife 2017 Feb 23;6:e23082. doi: 10.7554/eLife.23082) for example.

I also noticed a few typo that you may consider correcting :

  • line 307 there is the end of a sentence betwween Figure 1 and the legend;
  • line 316 the term HUFA is not defined and does not appear in the abbreviations list.

Author Response

Thank you for taking the time to review our article. We are grateful for your feedback. We have considered your thoughtful comments and made some corrections and clarifications accordingly. We have revised our manuscript and attached our point-by-point responses to each of your comments as follows

  1. The authors do not mention the potential risks, when speaking of dietary supplementation with PUFAs, of oxidation, and the need to supplement also with vitamin E, a point that could have impacts on ion channels activity in pathophysiological conditions.

Answer: Thank you for your thoughtful comment. We have briefly included a discussion of potential risks FA diet supplements will have (Line 54~510):

This suggests FAs be a putative sperm ion channel modulator to overcome sperm motility associated abnormalities. However, when considering dietary FAs as potential ion channel modulators, care should be taken to address the other aspects that may arise from fatty acid metabolism. This includes minimizing oxidative stress (i.e., ROS) generated in the FA metabolism process. Such adverse effects could be overcome by supplementing together with antioxidant vitamins (Vitamin E) [133].

  1. The study of sperm ion channels has been slowed down by the technical limitation regarding the use of patch-clamp on these very special cells. This point is not discussed, and I believe it is really an issue on this topic.

Answer:Thank you for your suggestion. We have included in the revised manuscript about the technical limitations as follow in Line 488~491.

In summary, sperm ion channels and their pharmacological modulations are relatively less explored due to many practical difficulties such as patch clamping technique requiring highly technical skills and processing a motile cell like sperm is more difficult than other cell types [132].

  1. I also noticed a few typos that you may consider correcting :

line 307 there is the end of a sentence between Figure 1 and the legend;

line 316 the term HUFA is not defined and does not appear in the abbreviations list.

Answer:Thank you for your comment and we apologize for the typos and formatting issues that were present in the manuscript. We have thoroughly reviewed and corrected them in the revised version of the manuscript.

Reviewer 2 Report

In this review, the authors summarized the modulation of fatty acids on human sperm ion channels. Different ion channels, such as calcium, potassium, proton, sodium and chloride channels, are important for human spermatozoan motility. Fatty acids affect human spermatozoan motility by modulating ion channels. Polyunsaturated fatty acids were shown to modulate sperm ion channels to increase sperm motility. On the contrary, saturated and monounsaturated fatty acids have little or no effects on the activity of ion channels. Basically, the manuscript is well-written and would be interesting to a wide range of readers. However, I might have some suggestions below.

  1. Line 113, what does (long lasting) mean?
  2. Some references are missing. Line 124, 16, 170, 231,
  3. Many studies on PUFAs’ effects on Kv7.1 and other channels should be mentioned. (DOI: 10.7554/eLife.51453, doi.org/10.3389/fphys.2017.00043)

Author Response

Thank you for taking the time to review our article. We highly appreciate your valuable comments in improving our manuscript and its quality. After considering the suggestions and comments,  we have revised our manuscript and attached our point-by-point responses to each of your comments as follows.

  1. Line 113, what does (long lasting) mean?

Answer: Thank you for your comment. We have removed the text fragment to avoid any confusions in the revised manuscript.

  1. Some references are missing. Line 124, 16, 170, 231

Answer: Thank you for pointing out the missing references. We have included the missing references in line 124, 170 and 231.

  1. Many studies on PUFAs’ effects on Kv7.1 and other channels should be mentioned.

Answer: Thank you for your suggestion. We have incorporated suggested facts in the revised manuscript to provide more complete overview.   (Line 429~432)

In addition to this, Elinder et al [119] reviewed different effects of PUFAs on different types of voltage-gated potassium channels, where some of the potassium channels (KV7, Kv10-12, and Kca) resulted in an increased amplitude of current in the presence of the PUFAs.